# Updated estimation of cutaneous leishmaniasis incubation period in French Guiana

Romain Blaizot[1,2,3]*, Albin Fontaine[4,5,6], Magalie Demar[3,7], François Delon[8,9], Albane de Bonet d'Oleon[10], Aurélie Mayet[9,10], Franck de Laval[9,10], Vincent Pommier de Santi[4,10], Sébastien Briolant[4,5,6]

**1** Cayenne Hospital Center, Dermatology Department, Cayenne, French Guiana, **2** UMR TBIP 1019 Tropical Biomes and Immunophysiopathology, University of French Guiana, Cayenne, French Guiana, **3** National Reference Center for Leishmania, Cayenne, French Guiana, **4** Aix Marseille Université, Institut de Recherche pour le Développement (IRD), Assistance Publique-Hôpitaux de Marseille (AP-HM), Service de Santé des Armées (SSA), Vecteurs–Infections Tropicales et Méditerranéennes (VITROME), Marseille, France, **5** Institut Hospitalo-Universitaire (IHU)–Méditerranée Infection, Marseille, France, **6** Unité de Parasitologie Entomologie, Département de Microbiologie et Maladies Infectieuses, Institut de Recherche Biomédicale des Armées (IRBA), Marseille, France, **7** Cayenne Hospital Center, Parasitology Laboratory, Cayenne, French Guiana, **8** Direction Interarmées du Service de Santé en Guyane, Cayenne, Guyane, **9** Aix-Marseille University, INSERM, IRD, SESSTIM (Economic and Social Sciences, Health Systems, and Medical Informatics), Marseille, France, **10** SSA (French Military Health Service), CESPA (French Armed Forces Center for Epidemiology and Public Health), Marseille, France

* Blaizot.romain@gmail.com

**Data Availability Statement:** All relevant data are within the manuscript and its Supporting Information files.

## Abstract

### Background

The cutaneous leishmaniasis (CL) incubation period (IP) is defined as the time between parasite inoculation by sandfly bite and the onset of the first CL lesion. IP distribution is difficult to assess for CL because the date of exposure to an infectious bite cannot be accurately determined in endemic areas. IP current estimates for CL range from 14 days to several months with a median around 30–60 days, as established by a few previous studies in both New and Old Worlds.

### Methodology

We estimated CL incubation period distribution using time-to-event models adapted to interval-censored data based on declared date of travels from symptomatic military personnel living in non-endemic areas that were exposed during their short stays in French Guiana (FG) between January 2001 and December 2021.

### Principal findings

A total of 180 patients were included, of which 176 were men (97.8%), with a median age of 26 years. When recorded, the parasite species was always *Leishmania guyanensis* (31/180, 17.2%). The main periods of CL diagnosis spread from November to January (84/180, 46.7%) and over March-April (54/180, 30.0%). The median IP was estimated at 26.2 days

**Funding:** The author(s) received no specific funding for this work.

**Competing interests:** The authors have declared that no competing interests exist.

(95% Credible Level, 23.8–28.7 days) using a Bayesian accelerated failure-time regression model. Estimated IP did not exceed 62.1 days (95% CI, 56–69.8 days) in 95% of cases (95th percentile). Age, gender, lesion number, lesion evolution and infection date did not significantly modify the IP. However, disseminated CL was significantly associated with a 2.8-fold shortening of IP.

## Conclusions

This work suggests that the CL IP distribution in French Guiana is shorter and more restricted than anticipated. As the incidence of CL in FG usually peaks in January and March, these findings suggest that patients are contaminated at the start of the rainy season.

## Author summary

Cutaneous leishmaniasis is a disease caused by a parasite transmitted to humans by the bites of sandflies. As these bites usually go unnoticed, it is not clear how long the incubation period (IP) lasts between the infectious bite and the appearance of skin lesions. In this study, we determined the IP distribution based on the dates of arrival, departure, and appearance of skin lesions from military personnel coming from non-endemic areas that were exposed to sandflies bites upon their arrival in French Guiana during the last 20 years. The median IP was 26 days with a 5th and 95th percentile of 7.9 and 62 days, respectively. These IPs are much shorter than those previously reported. Soldiers with disseminated leishmaniasis (more than 10 lesions on two body parts) had a median IP significantly shortened by 21.6 days. Considering our results and the fact that cutaneous leishmaniasis is mostly seen in January and March, people in French Guiana are most probably contaminated at the beginning of the rainy season, between December and February, and not during the dry season, as it was previously admitted.

## Introduction

Cutaneous leishmaniasis (CL) is a neglected tropical disease mostly found in the Americas, the Mediterranean basin, the Middle East and Central Asia [1]. Between half a million and a million new cases are diagnosed each year [1]. This protozoan disease is caused by parasites of the *Leishmania* genus, which are transmitted to humans through sandfly bites.

In French Guiana, about 200 human cases of CL are diagnosed each year, with the most frequent species causing this disease being *L. guyanensis* (85% of cases), followed by *L. braziliensis* (10%) and rarer species such as *L. lainsoni*, *L. amazonensis* and *L. naiffi* [2–4]. Most cases occur at the beginning of the short rainy season (November-February), with another peak in March-April [3]. The most affected population is represented by Brazilian gold miners working illegally in the forest hinterland [3]. The second one is represented by military personnel, mostly engaged in the fight against illegal gold mining [3]. Soldiers' exposure to *Leishmania* parasites in FG, whether sporadic or during outbreaks, has been well reported [5–9].

The Incubation Period (IP) is defined as the time between parasite inoculation by sandfly bite and the onset of the first CL lesion [10]. Knowledge of the IP is important to allow for a good diagnosis and understand the dynamics of CL cases in different settings. Previous

cutaneous leishmaniasis IP estimates rely on few studies. In French Guiana, Nacher and colleagues estimated a cutaneous leishmaniasis IP median ranging from 42 to 49 days depending on the dry or rainy season [11]. Still regarding the Amazon basin area, Oré and colleagues estimated a median IP of 56 days with a minimum and maximum of 14 days and 252 days, respectively [12]. To the best of our knowledge, the only study specifically designed to determine IP among many individuals was performed in Tunisia between 2015 and 2019 [13]. In this study, Aoun *et al.* determined IP in patients dwelling in areas that were non-endemic for *Leishmania major* Zoonotic CL (ZCL) and were infected during trips to local foci of ZCL [13]. A median IP of 35 days was found, with values ranging from 7 to 147 days. However, these findings may not apply to South America, where *Leishmania* species, transmission cycles and human epidemiology are very different from Northern Africa [3,14]. General guides or book chapters on cutaneous leishmaniasis report IP ranging from 14 to 56 days with exceptional cases as long as three years [10,15]. All these estimates were either based on time intervals between consultations and declared dates of probable infection or resulted from a very strict data filtering to select trips in endemic areas shorter than a week. Time of infections were thus considered as directly observed events without censoring.

IP is usually very hard to determine in endemic areas where infections occur year-long, and the accurate time of exposure cannot be easily established. Besides, prospective cohorts where patients would be examined until the occurrence of clinical signs are not straightforward to implement. Military personnel represent an ideal population to study CL IP. Indeed, soldiers sent to French Guiana are coming from non-endemic areas of mainland France, stay in FG for a few weeks or months and are prospectively followed with great accuracy, as part of the epidemiological surveillance system in French armed forces. Here, we determined cutaneous leishmaniasis incubation periods distribution based on dates of travel and symptom onsets from military personnels that traveled from non-endemic areas to French Guiana using interval-censored time-to-event models [16]. Our updated determination of CL IP distribution shed a new light in determining the period at risk of CL, with important consequences for prevention means.

## Methods

### Ethics statement

Concerning ethical considerations, all collected data were part of the French Armed Forces medical surveillance system. Verbal consent was obtained for all participants. According to French law, ethical approval was not required for mandatory epidemiological surveillance. All collected date were anonymized. The database was declared to the CNIL (*Commission Nationale Informatique et Libertés*), the French data protection authority (Registration no. 2035221), on February 15, 2017. This surveillance activity was carried out in accordance with the General Data Protection Regulation (EU Regulation 2016/679).

The inclusion criteria for patients with a record of cutaneous leishmaniasis were as follows: (1) have a confirmed CL skin lesion (typical wet ulcer with raised borders) and at least one parasitological test among smear, culture on skin biopsy and PCR on skin biopsy or swab [2,17] after their return from an area with a documented *Leishmania spp.* transmission, (2) have declared symptoms onset date (denoted here as S) and travel dates (denoted here as A and D for arrival in and departure from the area with *Leishmania spp.* transmission, respectively) (Fig 1). The departure date was not considered as mandatory for people that had developed symptoms during their stay in the transmission area.

All information about CL cases was collected during 2001–2021, in France or in French Guiana, by the military physicians by using a mandatory specific form. This form includes

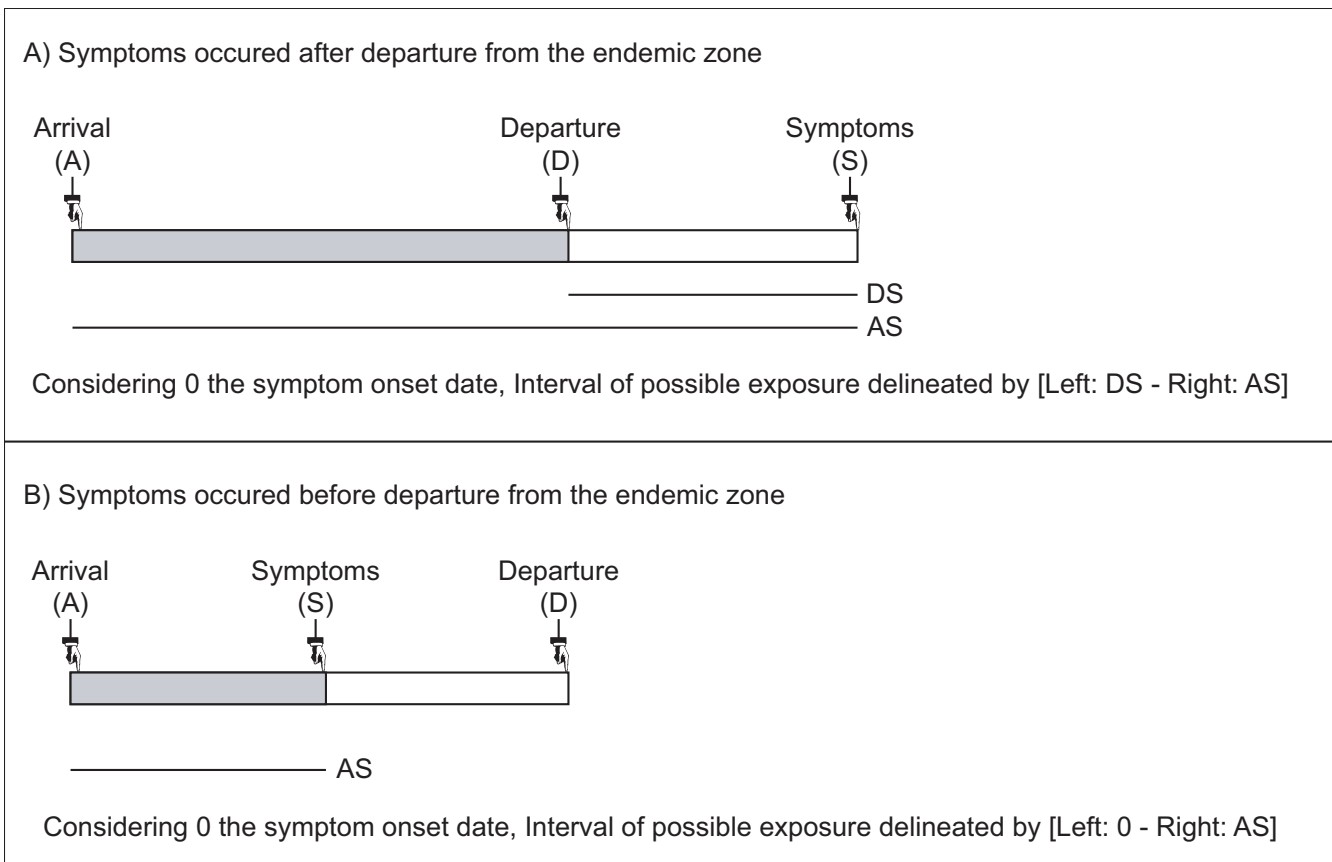

**Fig 1. Incubation Period (IP) distribution determination by interval censoring, using time of arrival (A), departure (D) from the area with a known *Leishmania* transmission and onset of symptoms (S).**

sociodemographic information (age, gender, military unit), clinical signs (symptoms onset date, lesions number, localization, dimension and dissemination) and outcomes (cure or failure), diagnostic tests (smear, culture on skin biopsy and PCR on skin biopsy or swab), treatment data (type, date, mode of administration and side effects) and travel data (location, arrival and departure dates). Species were determined from 2014 to 2019 with a PCR-RFLP technique targeting a 615-bp *Leishmania* genomic region of the RNA Pol II gene on skin biopsies [2]. After 2019, CL molecular diagnostic was performed using SYBR green-based real time PCR targeting the Hsp70 gene, followed by Sanger sequencing on swab samplings, according to previously published studies of the Cayenne Hospital Center [17]. Of note, all military personnel deployed to FG benefit from a consultation with a general practitioner on an annual basis and prior to any deployment.

We assumed that exposure to *Leishmania spp.* only occurred via the bite of an infectious sandfly in areas with a known exposure potential to phlebotomine sand flies and *Leishmania* transmission, which were mainly training fields such as the "Centre d'entraînement en forêt équatoriale (CEFE)" or deep forest during anti-illegal gold mining operations. The moment of infection via a sandfly bite is not directly observable, whereas the end event, the onset of symptoms, is always known for symptomatic patients. Time of infection events were thus subject to single interval censoring. For travelers who experienced symptoms after returning from travel, the exposure interval was delineated by the arrival date and the departure date in and from

areas with *Leishmania* transmission (AD period). For travelers who experienced illness during their travel in areas with *Leishmania* transmission, the exposure interval was delineated by the arrival date and the symptom onset date (AS period) (Fig 1).

We assumed that the IP and the censoring times were independent (*ie*, that the distribution of the inspection times was independent of the event time of interest). The icfit function from the interval (version 1.1.0.1) R package [18] was used to calculate non-parametric maximum likelihood estimates (NPMLEs) of the IP distribution (generalization of the Kaplan-Meier estimate) with a modified bootstrap confidence interval (CI) method. Estimates were calculated over a set of time intervals called Turnbull intervals, which represent the innermost intervals over a group of individuals in which NPMLE can change. The non-parametric log-rank test, as implemented in the ictest function with default parameters, was used to assess the statistical independence of the duration of IP with age, gender, lesion number, lesion dissemination, evolution, and the date of infection. Disseminated CL was defined by the presence of more than 10 lesions on two non-contiguous body parts [19]. Age, lesion number and time of infection parameters were categorized in two or three factors prior to analysis (S1 and S2 Files).

Age was grouped in two categories based on the median age of the cohort. The impact of one lesion versus several lesions on IP was assessed. A Bayesian accelerated failure-time regression model (AFT) with a gamma error distribution was fitted to the data using the ic_bayes function from the icenReg R package [20] using flat priors, 4 Markov chain Monte Carlo (MCMC) chains to run, 10000 samples, and 1000 samples discarded for burn in. The gamma distribution was chosen over other distributions based on visual comparison between several parametric baseline distributions and the semi-parametric distribution, as implemented in the diag_baseline function (S3A Fig).

The same model was used to model the lesion dissemination impact on IP. NPMLE is a generalization of the Kaplan Meier curves that allows for interval censoring. This method is not constrained by conditions of application, but only give survival estimates as time intervals (*i.e.* the probability of occurrence of an event is the same over the interval). Accelerated failure time (AFT) parametric models have the asset to provide a smooth survival function and to constrain the model flexibility, which thereby give more accurate estimation under the assumption that the chosen survival function distribution fits the data. Lesion dissemination effect was assumed to be consistent throughout time post infection. In addition to the package specified above, the survival (version 2.41.3) [21] and flexsurv (version 1.1) [22] R packages were used to analyze the data in the R statistical environment.

## Results

In total, our study population was constituted of 180 traveling soldiers meeting the inclusion criteria, as presented in the PRISMA Flowchart (**Fig 2**).

Regarding the general characteristics of this cohort, the median age was 26 years (Interquartile (IQR) 23–31), with minimum and maximum ages of 19 and 52 years. Almost all patients were men (176/180, 97.8%). The most frequent place of exposure was the CEFE on the Eastern coastal region (83 patients, 46.1%) The main periods of CL diagnosis spread from November to January (84/180, 46.7%) and over March and April (54/180, 30.0%). The infecting species was isolated in only 31/180 cases (17.2%) and was always *L. guyanensis* (31/31, 100%). Twelve cases of disseminated CL were recorded, including 8 in 2014 and 4 in 2020. The clinical characteristics of the study population are presented in Table 1.

We assessed CL incubation period distribution using dates of travel and symptom onsets (S2 Fig).

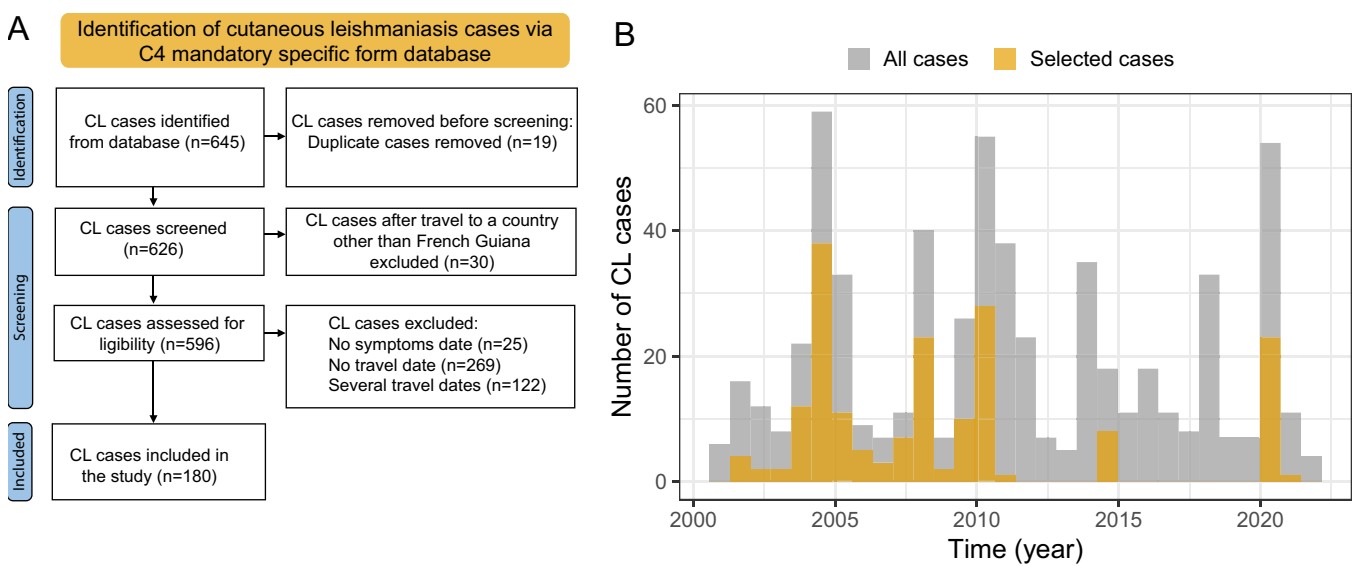

**Fig 2. (A) Prisma 2020 [23] Flow diagram of the database screening process and (B) histogram representing leishmaniasis cases from our database and the total number of recorded CL from 2000 to 2020.**

A median time of 25–26 days (95% confidence interval [CI], 20–30 days) from infection to symptoms onset was estimated using nonparametric maximum likelihood estimators (NPMLE) (S3 Fig).

The median IP was refined to 26.2 days (95% CI, 23.8–28.7 days) when using Bayesian AFT regression (Fig 3).

The shape and scale of the gamma distribution were estimated to 1.075 (mean) ± 0.1197 (SD) and 2.31 ± 0.127, respectively (S3B and S3C Fig).

The estimated IP was 7.8 days (95% CI, 6.2–9.8 days) in 5% of cases (5th percentile) and did not exceed 62.1 days (95% CI, 56–69.8 days) in 95% of cases (95th percentile), according to Bayesian AFT regression. The median number of lesions was 1 (IQR 1–2) with a maximum on 15. The length of IP was not significantly impacted by the number of lesions (one *vs* several, log-rank test p = 0.52). Age, gender and lesion evolution, were not shown to significantly impact the duration of *Leishmaniasis* incubation (log-rank test p>0.05).

On the other hand, a deceleration factor of 0.36 was estimated using Bayesian AFT regression for patients experiencing lesion dissemination relative to patients not experiencing lesion dissemination. In other words, keeping all other factors constant, the expected IP of patients experiencing a lesion dissemination decreased by 2.8 fold as compared to patients not experiencing lesion dissemination.

The median time estimates of IP was 12.2 days (95% CL, 8.2–17.1 days) for patients experiencing lesion dissemination, as compared to 33.8 days (95% CI, 25.8–43.1 days) for patients without lesion dissemination (Fig 4).

Of note, estimations of CL IP was not influenced by the date of data collection (S4 Fig).

## Discussion

CL was repeatedly reported to occur after an extended incubation period that can last from few weeks to several months after an exposure to the bite of an infectious sand flies. A lack of standardization in (*i*) the IP measurement protocols, (*ii*) study localization (Old *vs* New World), or the studied population characteristics across previous studies manifested into a

**Table 1. General characteristics of soldiers with proven cutaneous leishmaniasis and available dates of arrival and departure, French Guiana, 2001–2021.**

| Variable | Number of patients | % |
|---|---|---|
| **Age** | | |
| ≤ 25 | 94 | 52.2 |
| >25 | 86 | 47.8 |
| **Sex** | | |
| Men | 176 | 97.8 |
| Women | 3 | 1.7 |
| N/A | 1 | 0.5 |
| **Microscopic examination** | | |
| Positive | 166 | 92.2 |
| Negative | 14 | 7.8 |
| **Culture** | | |
| Performed | 70 | 38.9 |
| Positive | 51 | 72.9 (51/70) |
| **Species identification** | | |
| Performed | 31 | 17.2 |
| *L. guyanensis* | 31 | 100 (31/31) |
| **Number of lesions** | | |
| Median | 1 | 1–2 (Q1-Q3) |
| Minimum | 1 | 0 |
| Maximum | 15 | |
| **Mean lesion size (smallest lesion)** | | |
| **Smallest lesion** | 14.7 | |
| **Biggest lesion** | 17.7 | |
| **Dissemination*** | 12 | 6.7 |
| **Evolution after pentamidine** | | |
| Failure | 18 | 14.4 |
| Success | 116 | 85.6 (116/134) |
| N/A | 46 | |
| **Exposition area** | | |
| CEFE training center (Regina) | 83 | 46.1 |
| Others | 117 | 53.9 |

*recorded from 2014 onwards

wide range of median IP estimates in the literature. Using 92 cases, Aoun and colleagues estimated a median IP of 35 days with a range from 7 to 147 days in Tunisia using 92 cases. These 92 samples resulted from a very strict data filtering to select trips in endemic areas shorter than a week, in order to infer IP with an accuracy of one week. It allowed the authors to estimate an IP distribution by considering IP as directly observed continuous data [13]. Using a similar method, Oré and colleagues estimated a median IP of 56 days with min 14 days and max 252 days in the Amazon basin [12]. Here, we assessed CL IP distribution using time-to-event models adapted to interval-censored data based on declared date of travels from symptomatic soldiers returning from areas with active leishmaniasis transmission. Our method considers the uncertainty of the time to observation (censoring) and infers probabilities of observing an event at a given time without the need to select only for short periods of exposure.

The main result of this study is an estimated median (26 days) and a 95th percentile (62.1 days) of the IP distribution that was considerably shorter than previous estimates. Given this

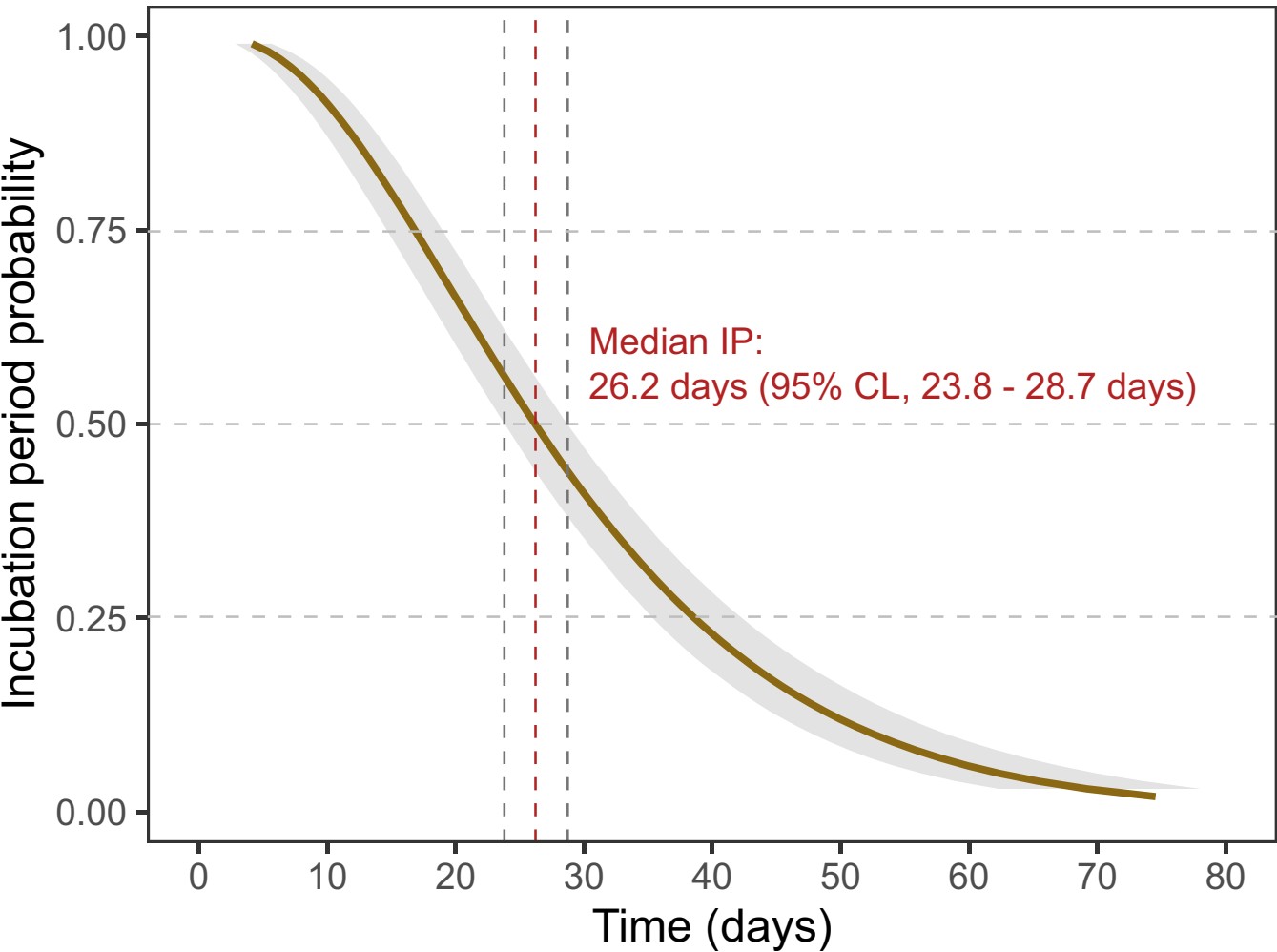

**Fig 3. Probabilities of cutaneous leishmaniasis incubation period (IP) over time post-infection calculated by interval-censored survival analyses using declared traveling periods of symptomatic patients returning from areas with *Leishmania* transmission.** Graphs represent the probability to be in the disease incubation stage (*i.e.* before symptom onset) over time post infection. An AFT with a gamma error distribution was fitted with a Bayesian Accelerated Failure Time model for interval censored data with flat priors to provide estimates with 95% Credible Levels.

very short delay between exposure and onset of symptoms and given that diagnosis in French Guiana is often made within a month of the onset and usually peaks in January and March, this model suggests a maximum probability of sandfly bites at the start of the rainy season in December. These data are in line with the occurrence of CL outbreaks with many cases after a sequence of dry and wet conditions [9]. This is consistent with factors associated with sandflies development and behavior, as they are more likely to look for blood meals in cool, shady and humid conditions, which are also important for larval development [24,25]. On the other hand, drought can delay the larval development [26]. Therefore, a period of dry weather followed by a rainy period could afford optimal conditions for a sudden increase in CL cases. In a retrospective study of the epidemiology of CL in French Guiana over the last five years, Blaizot *et al.* [27] showed that the number of cases seemed to decrease both in very dry and very wet years, as if the proper sequence of the dry season followed by the sudden return of rains was paramount to the occurrence of CL in this territory [27]. However, these data must be strengthened by the combination of more precise climatic studies and entomological

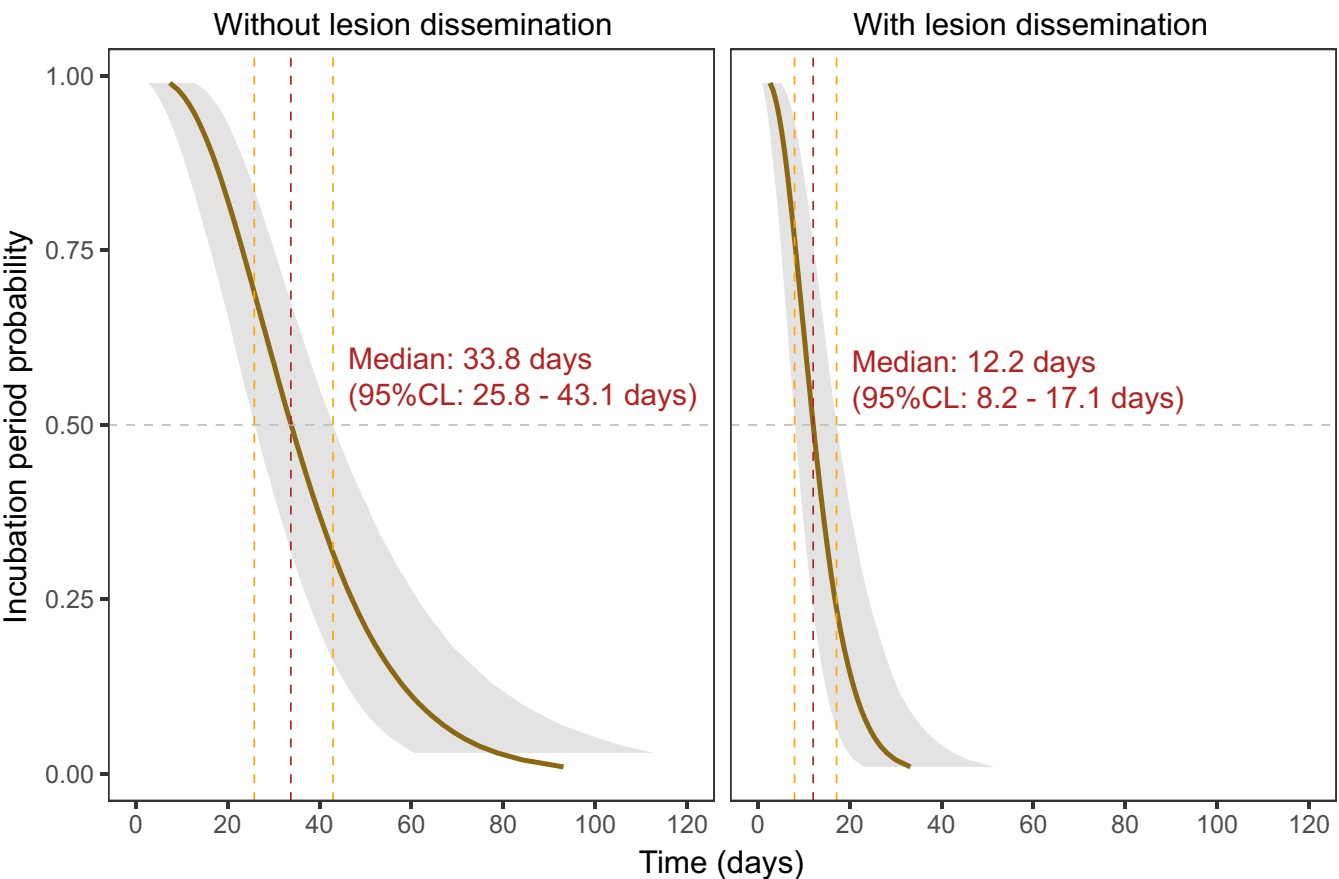

Lesion dissemination coefficient mean -1.029, SD: 0.2181, deceleration factor: exp(-1.029)= 0.36

**Fig 4. Probabilities of cutaneous leishmaniasis incubation period (IP) over time post-infection as a function of lesion dissemination.** Bayesian accelerated failure-time regression models (AFT) estimates are represented. Median estimates are represented with their 95% Credible Levels (CL). Light gray shading denotes 95% CL.

evaluation of sandfly behavior according to weather, particularly for *Lutzomyia umbratilis*, the main vector for *L. guyanensis* in FG.

Another finding of this study is the occurrence of a limited number of patients with very short incubations (*ie.* less than 8 days in 5% of cases). This has been reported both in French Guiana [11] and in other countries such as Tunisia [13] with incubations as short as a week. Therefore, a diagnosis of CL should not be ruled out because of a very short incubation, though this situation remains very rare (less than 5% of cases in our study). This finding is in line with our own experience, as a few cases of proven CL have been observed in patients arriving in FG just a week before the occurrence of symptoms. We found no association between the length of IP and age, gender or the number of lesions. Such associations may have been missed due to the homogenous character of our young, almost exclusively male study population.

On the other hand, it was shown that the dissemination of lesions was associated with a shortening of IP. Therefore, it seems that the potential for extension to multiple body parts is directly linked to the ability of CL to quickly induce symptoms. Disseminated CL (DCL) is a very specific form of CL where more than 10 lesions are found on more than two non-contiguous body parts [19]. These lesions are typical and contain few parasites. Often described with

*L. braziliensis*, DCL can also be caused by *L. guyanensis* [3], and must not be mistaken with diffuse cutaneous leishmaniasis, an anergic form usually caused by *L. amazonensis* [28]. A Brazilian study comparing clinical presentations and phylogenetic characteristics showed that specific strains of *L. braziliensis* were involved in DCL [29]. It is possible that similar genetic factors apply to *L. guyanensis* strains causing DCL. This would explain that a shortening of IP could be linked with dissemination but not with lesions number, as DCL represents a very specific form of disease, probably associated with specific strains which might also possess an ability to induce symptoms more quickly than others. Though genetic factors linked to the parasite seem to play an important role, further studies should also explore the impact of extrinsic factors such as host microbiota [30] or immune response [31] on the incubation period distribution.

This study has several limitations. Our findings in French Guiana may not apply to all settings of South America, and variations could be expected according to the local ecology of *Leishmania* species or human epidemiology. Associations between IP duration and clinical characteristics might also have been missed in our very homogenous population of young and healthy soldiers, with no comorbidities. The impact of the species involved could not be evaluated as it was determined in only 31 soldiers and cases of *L. braziliensis* might have been missed. Incubation in species other than *L. guyanensis* could be different. However, the characterization of species was not available throughout the entire study period, which represents a bias. It is also noteworthy that the IP in prime-infected patients such as these soldiers could be shorter than in people living in endemic areas, as previous CL infections are known to provide an acquired immunity [32].

Based on the assumption that CL IP lasts several months [11], the occurrence of most cases around January would be explained by exposures to sandfly bites occurring two months earlier, during the FG dry season. Assumptions have been made that outdoor activities are more frequent during the dry, pleasant season and would explain the annual peak of detected clinical cases two-three months later [11]. However, this can be litigated by the fact that during the 2020 CL outbreak occurring among military personnel in the Center for Equatorial Forest Training (CEFE), CL infections appeared in a few weeks after the arrival of trainees in the endemic area [9]. The outbreak also occurred in March, during a very rainy period, suggesting that intensive exposures to sandfly bites could be found out of the dry season.

We present here the first study specifically designed to determine CL IP distribution in South America. Our findings call for an update of the epidemiology of cutaneous leishmaniasis with important implications in the prevention measures against sandfly bites. Early symptoms after a risk of exposure to infectious sand flies should not be dismissed as differential diagnoses and exposing sandfly bites should be expected at the beginning of the rainy season, when protective and preventive measures should be enhanced. Further studies should also look for associations between specific *L. guyanensis* strains and the occurrence of DCL.

## Supporting information

**S1 File. R code to analyze interval censored time-to-event data and create visualizations.** The file provides code lines used to analyze the tabulated data provided in S2 File. (R)

**S2 File. Tabulated data providing sociodemographic information (age, gender), clinical signs (symptoms onset date, lesions number, dimension, dissemination and evolution), diagnostic tests (culture on skin biopsy or PCR on skin biopsy or swab) and travel data (location, arrival and departure dates).** (TXT)

**S1 Fig. Bayesian Accelerated Failure Time model coefficient estimation with the gamma baseline model describing the distribution of cutaneous leishmaniasis incubation period.** A- Diagnosis plot that displays the fit of the parametric gamma baseline model to Kaplan-Maier estimates. B- Summary table of the Bayesian Accelerated Failure Time model. C- Trace plots of Markov Chain Monte Carlo (MCMC) output and marginal density estimates for each gamma baseline distribution coefficient estimated with a Bayesian Accelerated Failure Time model for interval censored data with flat priors.
(EPS)

**S2 Fig. Representation of the length of travel periods as a function of symptom onset date in the retrospective cohort study.** Date of symptoms is represented at time 0 with a vertical red line. Each horizontal line represents the travel period for each individual 4in the cohort, with an orange circle and green circle corresponding to the arrival to French Guiana and departure date. Infection with *Leishmania guyanensis* are represented with blue lines.
(EPS)

**S3 Fig. Non-parametric maximum likelihood estimates of cutaneous leishmaniasis incubation period (IP) over time post-infection calculated by interval-censored survival analyses using declared traveling periods of symptomatic patients returning from areas with *Leishmania* transmission.** Median non-parametric IP estimates are represented with 95% confidence intervals.
(EPS)

**S4 Fig. Probabilities of cutaneous leishmaniasis incubation period over time were not significantly influenced by time of occurrence of the disease.** Histogram representing leishmaniasis cases from the database over time with a color code assigned to infections occurring (i) before 2006, (ii) between 2006 and 2012 and (iii) after 2012. Non-parametric maximum likelihood estimates of cutaneous leishmaniasis incubation period (IP) over time post-infection are provided for each of the three database subsets.
(EPS)

## Author Contributions

**Conceptualization:** Romain Blaizot, Albin Fontaine, Sébastien Briolant.

**Formal analysis:** Romain Blaizot.

**Investigation:** Romain Blaizot, Albin Fontaine, Magalie Demar, François Delon, Albane de Bonet d'Oleon, Aurélie Mayet, Franck de Laval, Vincent Pommier de Santi, Sébastien Briolant.

**Methodology:** Sébastien Briolant.

**Writing – original draft:** Romain Blaizot.

**Writing – review & editing:** Albin Fontaine, François Delon, Aurélie Mayet, Franck de Laval, Vincent Pommier de Santi, Sébastien Briolant.

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
