## [Decision Letter · Decision Letter 0]

12 Dec 2022

Dear Dr Blaizot,

Thank you very much for submitting your manuscript "Cutaneous Leishmaniasis incubation period in French Guiana: a paradigm shift" for consideration at PLOS Neglected Tropical Diseases. As with all papers reviewed by the journal, your manuscript was reviewed by members of the editorial board and by several independent reviewers. In light of the reviews (below this email), we would like to invite the resubmission of a significantly-revised version that takes into account the reviewers' comments. 

Very interesting study addressing a poorly describe parameter of leishmania epidemiology. However, the data presented needs to be properly expanded to better understand what conclusions can be obtained from it. Also, the limited nature of the surveillance record data available prevents from making solid conclusions. A paradigm shift would not result only from this preliminary data, but would require a properly conducted validation study. And in general, more specific acknowledgement of limitations is needed, and their impact in the strength of the conclusions

We cannot make any decision about publication until we have seen the revised manuscript and your response to the reviewers' comments. Your revised manuscript is also likely to be sent to reviewers for further evaluation.

Sincerely,

Andres Guillermo Lescano, PhD, MHS

Guest Editor

Ricardo Fujiwara

Section Editor

Very interesting study addressing a poorly described parameter of leishmaniasis' epidemiology. However, the data presented needs to be properly expanded to better understand what conclusions can be obtained from it. Also, the limited nature of the surveillance record data available prevents from making solid conclusions. A paradigm shift would not result only from this preliminary data, but would require a properly conducted validation study. And in general, more specific acknowledgement of limitations is needed, and their impact in the strength of the conclusions, and the actual possibility of a “paradigm shift”.

Major comments:

There is insufficient description of the nature and contents of the surveillance records used, how they are collected, where data was recorded (in France, in French Guiana or both), how records were extracted for the study, etc. A standard PRISMA flowchart should accompany the manuscript to understand how much data was eliminated in order to reach the study sample. Also, how in the records it was determined that personnel where in leishmania endemic areas? A deployment in Cayenne city would count as being in endemic areas? How was this accounted for in the existing records? More description on what was done (and not done) is needed. How does this affect the accuracy of the incubation period estimates?

There is insufficient description of the actual results. No data is presented on the minimum and maximum incubation periods, nor the minimum or maximum travel period. No table with descriptive data is presented. Greater emphasis is placed in estimates from statistical calculations that in minimally descriptive data (how about an histogram of actual observed incubation periods?). This is at odds with statements about how long can the incubation period be, because no data is provided as reference in this case series. Apparently there were several cases that had incubations periods > 2 months, but it’s hard to understand that in the paper itself.

An important emphasis is placed in statistical methods, but the choices are not well justified (why NPMLEs? Why AFT regression over proportional hazards?). Some answers may be somewhat obvious but others are not. Two estimates are presented sometimes, which is the right one and why is the other presented? Again, no tables are presented with regression coefficients, confidence/credible intervals, etc. This affects the readability of the manuscript 

Given that the authors postulate that their observed IP are shorter than previously described in the literature, the immediate question is, how accurate is the surveillance system to detect leish cases with long incubation periods? Could they have been missed, resulting in underestimation of the median/mean IP? Could further deployments of military personal soon after French Guiana prevented the ability of the records to attribute exposure to their stay in French Guiana, missing infections with longer incubations periods? Supplementary Figure 1 shows that the vast majority of leish diagnoses were made before departure, suggesting that diagnoses made back. It is likely that cases with longer incubation periods could have been misdiagnosed back home, biasing estimations. Only a prospective study with long-follow up could properly address this issue.

The authors describe incubations periods in the literature as short as a week, but Supp Figure 1 includes multiple cases with incubation periods shorter than five days, and even shorter. This is biologically very unlikely and suggests a potential problem in the recording of the symptom initiation or the confirmation of diagnoses, casting doubt on the short incubation period estimates. This issue needs to be recognized and address in the limitations of the study.

Time trends over 20 years and Leishmaniasis outbreaks, a very common situation in military deployments, can severely impact estimates. Outbreaks result in highly correlated events, violating the independence assumption of the tests used, and potentially altering both point estimates as well as confidence intervals. Time trends, potentially with fewer military deployments and less cases in recent years could lead to estimates based in older and potentially more error-prone data. A thorough analysis of the impact and role of these two factors in the results are needed.

I am not sure it’s accurate to present the proportion of Guyanensis cases as 30/180 (16.7%), as 30/30 samples were positive (100% positivity) and 16.7% only reflects the testing rate, not the positivity rate. However, it would be relevant to know whether more samples were tested and results were inconclusive. And, I imagine results mostly reflect recent years where species differentiation started to be performed, with obvious biases.

Reviewer's Responses to Questions

**Key Review Criteria Required for Acceptance?**

**Methods**

-Are the objectives of the study clearly articulated with a clear testable hypothesis stated?

-Is the study design appropriate to address the stated objectives?

-Is the population clearly described and appropriate for the hypothesis being tested?

-Is the sample size sufficient to ensure adequate power to address the hypothesis being tested?

-Were correct statistical analysis used to support conclusions?

-Are there concerns about ethical or regulatory requirements being met?

Reviewer #1: 1.Objective of the study is clear. 

2.Study design is appropriate. Regarding the pre-recruitment status of the study participants: were they screened for already existing CL lesions or travel history to CL endemic areas, prior to deployment to French Guiana? You may mention these in the manuscript or mention as limitations.

3.Please mention exclusion criteria when describing the study population.

4. Regarding the sample size: please clarify whether 180 is the whole study population or whether it was a selected sample. If it is a selected sample, please mention the sample size calculation and the sampling technique. 

5.Regarding statistical analysis: please include the justification for using non-parametric approaches for distribution identification and for assessing the independence of variables.

It will add clarity, if the reason behind using a Bayesian AFT regression model is included to the manuscript.

6. Ethical/regulatory requirements have been fulfilled.

Reviewer #2: Are the objectives of the study clearly articulated with a clear testable hypothesis stated?

Answer: No

-Is the study design appropriate to address the stated objectives?

Answer: No

-Is the population clearly described and appropriate for the hypothesis being tested?

Answer: In part, yes. However, some information is lacking regarding the number of trips to the endemic area.

-Is the sample size sufficient to ensure adequate power to address the hypothesis being tested?

Answer: No

-Were correct statistical analysis used to support conclusions?

Answer: Yes

-Are there concerns about ethical or regulatory requirements being met?

Answer: No

**Results**

-Does the analysis presented match the analysis plan?

-Are the results clearly and completely presented?

-Are the figures (Tables, Images) of sufficient quality for clarity?

Reviewer #1: 1. It is mentioned in the analysis plan that IP distribution identification will be done with parameter estimation. However, result of the proposed distribution identification is not clearly given in the results section.

2. It is mentioned ‘An AFT with a gamma error distribution was fit with a Bayesian regression model to provide Bayesian estimates.’. Interpretation of the regression estimates relating to the application has not been presented in the manuscript. 

3. To add clarity and value to the manuscript, please include the equation of the final fitted AFT regression model.

4. Figures are of sufficient quality.

Reviewer #2: Does the analysis presented match the analysis plan?

Answer: Yes

Os resultados são apresentados de forma clara e completa?

Answer: No

Are the figures (Tables, Images) of sufficient quality for clarity?

Answer: Yes

**Conclusions**

-Are the conclusions supported by the data presented?

-Are the limitations of analysis clearly described?

-Do the authors discuss how these data can be helpful to advance our understanding of the topic under study?

-Is public health relevance addressed?

Reviewer #1: 1. Including the equation of the final fitted AFT regression model to the manuscript will give sufficient evidence for the conclusions derived.

2. Limitations of the analysis have been described.

3. Authors have briefly discussed how the study data will be helpful and the public health relevance.

Reviewer #2: -Are the conclusions supported by the data presented?

Answer: No

-Are the limitations of analysis clearly described?

Answer: Yes

-Do the authors discuss how these data can be helpful to advance our understanding of the topic under study?

Answer: Yes

-Is public health relevance addressed?

Answer: Yes

**Editorial and Data Presentation Modifications?**

Reviewer #1: Overall, the manuscript is well-written, clear and easy to understand. I suggest the following:

Line 1: Upper case ‘L’ in Leishmaniasis change to lower case

Line 27: Upper case ‘L’ in Leishmaniasis change to lower case

Line 57: Upper case ‘L’ in Leishmaniasis change to lower case

Line 71: Upper case ‘L’ in Leishmaniasis change to lower case

Line 74: sandflies change to sandfly

Line 86: et al needs to be in italics

Line 97: Soldiers change to Soldiers’

Line 209: Upper case ‘L’ in Leishmaniasis change to lower case

Line 221: Upper case ‘L’ in Leishmaniasis change to lower case

Reviewer #2: (No Response)

**Summary and General Comments**

Reviewer #1: Interesting study findings and will be useful for updated understanding of CL. Manuscript needs the revisions mentioned earlier.

Reviewer #2: In this masnucript the authors evaluated the incubation period of cutaneous leishmaniasis in a cohort of French soldiers who developed cutaneous leishmaniasis after a period of stay in areas of transmission of Leishmania, in French Guiana. The topic explored is of interest to the community, however the aforementioned manuscript has many flaws for assertive conclusions to be explored. Below is the review:

1) Introduction

Page 5, lines 80-82. The authors refer to the ancient experiment on the use of live vaccines to induce preventive sores. This information is out of context, because in this condition there is no participation of the saliva of the vector, which has an important role to recruit cells and also facilitate the entry of the parasite into the host cell.

2) Methods

Page 8, line 124. The authors cited CL skin lesion, however they not mentioned the type and characterisitic of the lesion. So I ask: What was the characteristic lesion? This information is important, because the beginning of the LC lesion is a papule, which evolves into a nodule and then ulcerates, so according to the lesion characterized, there may be a difference in the interpretation of the incubation period.

Regards trip to endemic area: Were only patients included who had their first trip to FG? Or were there some patients like more than one trip?

3) Results. According to the authors, 30 from 180 samples were characterized as L. guyanensis. This sampling is too small to say that L. guyanensis is the main species, because in different areas of the FG there may be circulation of other species, such as L. braziliensis. The authors also did not inform how the species was characterized.

Ragarding the distribution of the patients over the 20 years of studies, some points need to become clearer.

How many patients had the disseminated cutaneous form? Was there a difference between the years of inclusion of the patients? The authors evaluated a period of 20 years. It is important to demonstrate the average number of cases per year.

4) Discussion. The authors suggest that the strains of L. guynensis causing LCL are different from the strains causing LCL but this has not been demonstrated and there is no evidence for this species.

Another important point is about the development of lesions in prime-infected people and those residing in transmission areas, which are frequently exposed to the bite of the vector insect. I suggest include this in the discussion.

PLOS authors have the option to publish the peer review history of their article (what does this mean?). If published, this will include your full peer review and any attached files.

Reviewer #1: No

Reviewer #2: No
---

## [Decision Letter · Decision Letter 1]

12 May 2023

Dear Dr Blaizot,

Thank you very much for submitting your manuscript "Updated estimation of cutaneous leishmaniasis incubation period in French Guiana" for consideration at PLOS Neglected Tropical Diseases. As with all papers reviewed by the journal, your manuscript was reviewed by members of the editorial board and by several independent reviewers. The reviewers appreciated the attention to an important topic. Based on the reviews, we are likely to accept this manuscript for publication, providing that you modify the manuscript according to the review recommendations. 

The authors have done a great job in addressing nearly all comments of the reviewers. Please do address these minor issues

1) Include the PRISMA flowchart in the manuscript and not in the supplementary material

2) I would not consider the term "contamination" to describe exposure to sandfly bytes or exposure to CL

3) Your responses in the letter, particularly in the justification of methods used and decisions made, are important and informative. They clarify concerns of the reviewers that most lilkely will be concerns of readers as well. Please include them in the actual manuscript, You can summarize them a little bit if felt needed

4) Some descriptions of the methods are presented in the results when describing the findings. THey should be moved to the methods

Sincerely,

Andres Guillermo Lescano, PhD, MHS

Guest Editor

Ricardo Fujiwara

Section Editor

The authors have done a great job in addressing nearly all comments of the reviewers. Please do address these minor issues

1) Include the PRISMA flowchart in the manuscript and not in the supplementary material

2) I would not consider the term "contamination" to describe exposure to sandfly bytes or exposure to CL

3) Your responses in the letter, particularly in the justification of methods used and decisions made, are important and informative. They clarify concerns of the reviewers that most lilkely will be concerns of readers as well. Please include them in the actual manuscript, You can summarize them a little bit if felt needed

4) Some descriptions of the methods are presented in the results when describing the findings. THey should be moved to the methods

Reviewer's Responses to Questions

**Key Review Criteria Required for Acceptance?**

**Methods**

-Are the objectives of the study clearly articulated with a clear testable hypothesis stated?

-Is the study design appropriate to address the stated objectives?

-Is the population clearly described and appropriate for the hypothesis being tested?

-Is the sample size sufficient to ensure adequate power to address the hypothesis being tested?

-Were correct statistical analysis used to support conclusions?

-Are there concerns about ethical or regulatory requirements being met?

Reviewer #1: (No Response)

Reviewer #2: (No Response)

**Results**

-Does the analysis presented match the analysis plan?

-Are the results clearly and completely presented?

-Are the figures (Tables, Images) of sufficient quality for clarity?

Reviewer #1: (No Response)

Reviewer #2: (No Response)

**Conclusions**

-Are the conclusions supported by the data presented?

-Are the limitations of analysis clearly described?

-Do the authors discuss how these data can be helpful to advance our understanding of the topic under study?

-Is public health relevance addressed?

Reviewer #1: (No Response)

Reviewer #2: (No Response)

**Editorial and Data Presentation Modifications?**

Reviewer #1: (No Response)

Reviewer #2: (No Response)

**Summary and General Comments**

Reviewer #1: (No Response)

Reviewer #2: In this revised version, the authors have incorporated many of the suggestions made by the reviewers. The introduction is clearer and directed towards the objectives of the manuscript. Several paragraphs were modified and new information was added to support the authors' hypothesis. In the methodology, information was added referring to methods of determination of Leishmania species. Furthermore, the results are clearer and more robust. The discussion has been improved and some points have been clarified.

In this way, I accepted the modifications made by the authors and I consider that the manuscript can be accepted for publication in the current format.

PLOS authors have the option to publish the peer review history of their article (what does this mean?). If published, this will include your full peer review and any attached files.

Reviewer #1: No

Reviewer #2: Yes: Jose Angelo Lauletta Lindoso

Figure Files:

Data Requirements:

Reproducibility:

References

---

## [Editor Report · Decision Letter 2]

24 May 2023

Dear Dr Blaizot,

We are pleased to inform you that your manuscript 'Updated estimation of cutaneous leishmaniasis incubation period in French Guiana' has been provisionally accepted for publication in PLOS Neglected Tropical Diseases.

Best regards,

Andres Guillermo Lescano, PhD, MHS

Guest Editor

Ricardo Fujiwara

Section Editor

Despite adding the flowchart as part of the manuscript, a reference as a supplemental figure was left, potentially erroneously:

"as presented in the PRISMA Flowchart (Supplementary Fig S4 A)."

---

## [Editor Report · Acceptance letter]

11 Jun 2023

Dear Dr Blaizot,

We are delighted to inform you that your manuscript, "Updated estimation of cutaneous leishmaniasis incubation period in French Guiana," has been formally accepted for publication in PLOS Neglected Tropical Diseases.

Best regards,

Shaden Kamhawi

co-Editor-in-Chief

Paul Brindley

co-Editor-in-Chief
